# Exploring the Relationship Between Low Back Pain, Physical Activity, Posture, and Body Composition in Older Women

**DOI:** 10.3390/healthcare13091054

**Published:** 2025-05-03

**Authors:** Barbara Rosolek, Dan Iulian Alexe, Krystyna Gawlik, Elena Adelina Panaet, Ilie Mihai, Bogdan Alexandru Antohe, Anna Zwierzchowska

**Affiliations:** 1Institute of Sport Sciences, The Jerzy Kukuczka Academy of Physical Education in Katowice, 40-287 Katowice, Poland; b.rosolek@awf.katowice.pl; 2Department of Physical and Occupational Therapy, “Vasile Alecsandri” University of Bacău, 600115 Bacău, Romania; alexedaniulian@ub.ro; 3Bialska Academy of Applied Sciences John Paul II, 21-500 Biala Podlaska, Poland; k.m.gawlik@gmail.com; 4Department of Physical Education and Sport, National University of Science and Technology Politehnica Bucharest, Pitesti University Centre, 060042 Bucharest, Romania; ilie.mihai@upb.ro; 5Department of Physical Education and Adapted Physical Activity, The Jerzy Kukuczka Academy of Physical Education in Katowice, 40-065 Katowice, Poland; a.zwierzchowska@awf.katowice.pl

**Keywords:** low back pain, posture, body composition, aged woman, exercise

## Abstract

Background: Low back pain (LBP) is a widespread and disabling condition affecting many older adults. Methods: This study aimed to establish correlations between LBP, physical activity, body composition, and posture in 114 women (mean age: 67.6 ± 5.68 years). Using a cross-sectional study design, LBP was assessed using the Oswestry Disability Index (ODI). Physical activity (PA) was measured using a pedometer to count the steps taken. Spinal curvatures in the sagittal plane were examined with a Rippstein Plurimeter. Results: Significant correlations were found between ODI and waist circumference (WC) (F(1, 113) = 7.574, *p* = 0.007), body mass index (BMI) (F(1, 113) = 11.660, *p* = 0.001), total fat (TF) (F(1, 113) = 8.806, *p* = 0.004, R^2^ = 0.072), and total steps (F(1, 113) = 4.446, *p* = 0.037). No associations were found between ODI and hip circumference (HC) (F(1, 113) = 6.257, *p* = 0.014, R^2^ = 0.52), waist-to-hip ratio (WHR) (F(1, 113) = 6.342, *p* = 0.013, R^2^ = 0.053), thoracic kyphosis (THK) (F(1, 113) = 0.290, *p* = 0.591, R^2^ = 0.003), or lumbar lordosis angle (LLA) (F(1, 113) = 0.290, *p* = 0.591, R^2^ = 0.003). Conclusion: These results of the study findings reveal the multifactorial nature of LBP syndrome and highlight the connection between LBP and ODI, BMI, WC, and TF in older women. Additionally, we emphasize the importance of implementing further prevention and intervention strategies to manage the clinical manifestation of LBP in the geriatric population. Healthcare providers can better support this population’s well-being by focusing on targeted interventions.

## 1. Introduction

The aging process is a complex phenomenon that occurs in humans, frequently associated with physiological, psychological, and biomechanical changes that impact quality of life [1].

Low back pain (LBP) is a widespread and disabling condition, with 60–70% of the population suffering at least one episode of LBP during their lifetime worldwide [2,3]. Clinically, it is characterized by mild to severe pain, primarily in the lower back. During acute episodes, LBP disrupts daily life activities and leads to prolonged absences from work [4], creating socio-economic challenges [5]. LBP affects more women than men and is generally defined as a non-specific self-limiting process [6]. Older women are particularly vulnerable to chronic LBP due to postmenopausal hormone changes, which also contribute to decreased bone mineral density [7] and consequently may contribute to functional limitation and musculoskeletal conditions [8].

The causes of LBP are complex and multifaceted, varying from minor muscle strain to a complex systemic issue. Primarily, this condition could arise from a biomechanical limitation, produced by trauma, muscle imbalances, or posture, which leads to intervertebral disk and articular facet overloading [9].

This situation is often amplified by increasing weight, age, and body composition (increase in body fat or sarcopenia). Consequently, this might lead to changes in the spine’s physiological sagittal curvatures, which further intensify LBP [10,11].

With aging, the vertebral column undergoes structural changes, including shortening (due to intervertebral disk dehydration), increased rigidity, and backward curvatures. These changes can lead to impaired function of internal organs, reduced physical activity, increased pain perception, and a decline in overall mood. Furthermore, many elderly individuals with chronic LBP may develop kinesiophobia [12]. Characterized by a fear–avoidant response to movement in an attempt to prevent pain, this behavior can significantly reduce physical activity and negatively impact body composition in patients with lower back pain (LBP) [13,14].

A lower level of physical activity in the elderly population also contributes to increasing adiposity [15]. Body composition is assessed with body mass index (BMI), which is widely used as a great indicator of nutritional status and health risk [16].

The World Health Organization (WHO, 2020) recommends that individuals aged 65 and older should engage in moderate-intensity aerobic activity for at least 150–300 min two or more days per week. All indications of the WHO are based on the central definition of physical activity, which is defined as any movement of the body in general or part of it, produced by the skeletal muscle, with energy expenditure required [17].

Previous research has suggested that physical activity and body composition may play important roles in developing and progressing lower back pain [13,18]. The factors that might lead to increased back pain include sedentary lifestyles. Low physical activity levels accelerate involution changes related to human aging and problems with the motor organs. Muscle strength decreases while body composition and posture change, resulting in disturbed balance in the spinal column, pelvis, and lower limbs, which can hurt locomotion and cause back pain [19]. Studies have also emphasized improper sitting position as an important risk factor of low back pain syndrome [20,21]. While alterations of thoracic and lumbar vertebrae have been linked to back pain in younger populations, the evidence in older adults is limited and inconsistent [22,23].

We initiate our study with the central question, “What are the relationships between low back pain intensity, body composition, physical activity, and spinal posture in older women?” This study aimed to establish the correlations between low back pain and physical activity in relation to specific body composition and posture in women aged 60 years and older. By highlighting these connections, this study underscores the need for effective intervention strategies in geriatric healthcare that could enhance their quality of life.

## 2. Materials and Methods

### 2.1. Participants

This study utilized a cross-sectional study design to examine 114 retired women, aged 60 to 80 years (67.6 ± 5.68 years). They were participants in the Third Age University programs and were living in an urban area with over 100,000 people. Data were collected at a single point in time. The purposive sampling method was used to recruit study participants while applying the following inclusion criteria: age over 60 years, written consent to participate in this study, and successful completion of the test using pedometers. The exclusion criteria were comorbidities and functional limitation conditions. Informed consent was obtained from all subjects involved in this study.

The research protocol was approved by the Bioethics Committee at the Jerzy Kukuczka Academy of Physical Education in Katowice, Poland (No. 9 with addendum 29.10.2020) and met the ethical standards of the Declaration of Helsinki, 2013 [24].

### 2.2. Subjects’ Assessment

The examinations were performed using direct participant observation and a face-to-face diagnostic survey.

The Oswestry Disability Index (ODI) was used to assess low back pain. The questionnaire consisted of ten topics, with the maximum score being 50. The scale was used to quantify disability, and 0–4 indicated no disability, 5–14 indicated minimal disability, 15–24 indicated moderate disability, 25–34 indicated severe disability, and, finally, >35 indicated total disability [25].

Physical activity was evaluated using the objective device to count the number of steps (pedometer, Yamax Inc., Tokyo, Japan). Daily step count was consecutively recorded for 7 days (standards as presented by Tudor-Locke [26]).

Sagittal spinal curvatures were examined using the Rippstein plurimeter to measure the angles of the thoracic (THK) and lumbar (Ll) spine [27]. The Rippstein plurimeter is a tool used to evaluate the ranges of sagittal spine mobility. The result was represented by the two values of angular inclination, which were directly read from the device. Thoracic kyphosis angle (THK) was measured between TH1 (1st thoracic vertebra) and TH12 (12th thoracic vertebra), whereas lumbar lordosis angle (LLA) was measured between L5 (5th lumbar vertebra) and TH12 [28].

The anthropometric parameters were obtained by measuring body height (BH), waist circumference (WC), and hip circumference (HC). BH was measured with a portable stadiometer (Seca 213, Germany) following the procedure proposed by the *National Health and Nutrition Examination Survey (NHANES) Anthropometry Procedures Manual* [29], while WC and HC data were collected after applying the WHO STEP protocol [30]. A low-frequency bioelectrical impedance method (Tanita TBF-300M) was utilized to evaluate body mass (BM) and body fat percentage (FAT%). Body mass index (BMI in kg/m^2^) and waist-to-hip ratio (WHR) were calculated [31] (Table 1).

### 2.3. Statistical Analysis

This study utilized a cross-sectional study design to examine 114 women aged 60 to 80 (67.6 ± 5.68 years) who were retired and participants of the Third Age University programs living in an urban agglomeration with over 100,000 people. Data were collected at a single point in time.

The data were analyzed using IBS SPSS Statistics, V.25.

The minimum sample size of 89 participants was determined based on an a priori analysis using G*Power software (version 3.1.9.4, from the University of Dusseldorf, Germany). The analysis was conducted with the following parameters (effect size: 0.15, α = 0.05, β = 0.95, number of predictors = 1).

The normality assumption of the data was checked using the Kolmogorov–Smirnov test, and it was observed that the data did not meet the normality assumption. After the analysis, we decided to use linear regression for data analysis, since it does not require variables to be normally distributed. The results of the analysis are presented in Table 2.

## 3. Results

A linear regression analysis demonstrated a significant relationship between ODI and WC (F(1, 113) = 7.574, *p* = 0.007). However, the R^2^ value of 0.063 indicates that ODI explains only 6.3% of the variability in WC.

The ODI did not significantly correlate with HC (F(1, 113) = 6.257, *p* = 0.014, R^2^ = 0.52) or WHR (F(1, 113) = 6.342, *p* = 0.013, R^2^ = 0.053). A significant relationship was observed between ODI and BMI (F(1, 113) = 11.660, *p* = 0.001). The R^2^ value of 0.094 indicates a medium effect size of ODI on BMI (Figure 1).

A significant regression was found between ODI and FAT (F(1, 113) = 8.806, *p* = 0.004, R^2^ = 0.072), indicating that ODI explained approximately 7.2% of the variance in FAT.

A significant regression was found between ODI and total steps (F(1, 113) = 4.446, *p* = 0.037). The R^2^ was 0.038, indicating that ODI explained approximately 3.8% of the variance in total steps.

Further analysis indicated no significant relationship between ODI and thoracic kyphosis (F(1, 113) = 0.290, *p* = 0.591, R^2^ = 0.003) or lumbar lordosis (F(1, 113) = 0.290, *p* = 0.591, R^2^ = 0.003). The very low R^2^ values indicate that ODI has no meaningful explanatory power for these spinal curvature parameters in the studied population (Figure 2).

## 4. Discussion

LBP syndrome is considered an important factor in limiting people’s everyday activity, and its incidence is permanently increasing [32], especially in the geriatric population, where we can see changes in body posture and body composition [33].

While our regression analysis indicates that the ODI did not significantly affect the subjects’ THK (F(1, 113) = 0.290, *p* = 0.591, R^2^ = 0.003) or LLA (F(1, 113) = 0.290, *p* = 0.591, R^2^ = 0.003), previous studies found that postural changes occur in the elderly population. Anwajler demonstrated that THK depth gradually increased with age in the 6th, 7th, and 8th decades of life since the aging process leads to the modification of the body posture due to changes in active body-stabilizing muscles’ activity [34]. With aging, muscle strength is reduced, leading to changes in body posture. As a natural consequence, this imbalance is compensated for by an anterior shift of the center of gravity and an increase in the spine curvatures. Singh also found a significant increase in THK in older women, but no significant changes were made in the case of lumbar lordosis [35,36]. Some authors found that an increased THK affected disability, a more significant number of falls due to the shifted center of gravity [37,38,39,40], and deteriorated quality of life [41]. This adverse increase in THK observed with age more often affected female patients [42]. Changes in body posture may increase LBP, causing functional difficulties and affecting quality of life [43]. The relationship between lumbar lordotic curvature and LBP remains controversial. Even though the traditional perspective is associated with increased lordosis, current research found that a decreased LLA has a potential role as a contributing factor of LBP [44].

Age-related vertebral changes are well documented and form the basis of both clinical understanding and biomechanical models of postural adaptation. Chronic low back pain affects the somatosensory integration of the inputs received from the proprioception signaling of the lumbar region. Further, the central processing is affected, and consequently, dysfunctional motor control adaptation appears [45]. Perturbation of the trunk control can also increase mechanical loading on the vertebral tissues [46]. According to Panjabi’s model of spinal stability, failure of the interplay of osteo-ligamentous spinal structure, active structure, and control systems can make the vertebral column unstable [47]. In elderly individuals, all these components are affected by the degenerative process, contributing to the modification of thoracic and lumbar curvature [48,49].

Although postural variables (THK and LLA) were not statistically significant in the analysis, we considered it relevant to keep and discuss them due to the important clinical implications they may have in the geriatric population. Thus, integrating these variables into the analysis contributes to a more comprehensive understanding of the biomechanical and functional context of low back pain among older people.

Another biological factor that could be associated with non-specific LBP is insufficient strength and endurance of the muscles. In chronic cases, LBP is more related to biological deconditioning, which includes the musculoskeletal system and physiological or cardiovascular aspects [50]. According to clinicians, LBP syndromes create discomfort, limitations of activity in everyday life, and, consequently, modification in body composition.

The results of this study demonstrated an association between PA, characteristics of body composition, and the incidence of disability defined by LBP. We observed statistically significant correlations between the scores of the ODI and WC (R^2^ = 0.063, *p* = 0.007), HC (R^2^ = 0.052, *p* = 0.014), WHR (R^2^ = 0.053, *p* < 0.05), BMI (R^2^ = 0.094, *p* = 0.001), and TF (R^2^ = 0.072, *p* = 0.004). Our empirical results are consistent with the conclusions of many authors [10,11,13,14]. However, each of these studies identifies only one of the factors as significantly related to ODI (physique, attitude, exercise, or condition). We conducted an analysis of four complementary factors that might affect the ODI (body composition, build, posture, and physical activity) in a specific group of older women.

Given the specific nature of the subjects that we had in this research (active, elderly women), the transposition of the results obtained to the general population may not be totally objective. Several important factors were not addressed in this paper and might deserve further analysis in future research. These factors are gender differences [51], cultural environment [52], cognitive health [53], and psychological factors [54]. However, analyzing the results obtained in our research, in which ODI, WC, HC, WHR, BMI, and TF were correlated with low back pain, with caution, we can state that people that are sedentary and have worse physical status may exhibit greater lower back pain.

The linear regression analysis demonstrated a weak relationship between ODI and WC (F(1, 113) = 7.574, *p* = 0.007). However, the R^2^ value of 0.063 indicates that ODI explains only 6.3% of the variability in WC. The ODI did not significantly correlate with HC (F(1, 113) = 6.257, *p* = 0.014, R^2^ = 0.52) or WHR (F(1, 113) = 6.342, *p* = 0.013, R^2^ = 0.053). The outcomes from the study of You et al. [55] suggest that subjects with an increased WHR are susceptible to developing LBP. Trujillo et al. [56] confirms that in a large study sample, the subjects with greater hip and WC experienced severe pain.

A weak relationship was observed between ODI and BMI (F(1, 113) = 11.660, *p* = 0.001). The R^2^ value of 0.094 indicates a medium effect size of ODI on BMI. Mintarjo [36] identified being overweight as a potential risk factor for degenerative disc pathologies in a similar study design. The study results indicate that BMI significantly affects LBP, with a chi-squared test value of 0.015 (*p* < 0.05). Furthermore, the findings reveal that subjects with an increased BMI are 6.089 times more likely to develop LBP.

A weak regression was found between ODI and TF (F(1, 113) = 8.806, *p* = 0.004, R^2^ = 0.072), indicating that ODI explained approximately 7.2% of the variance in TF. The correlation between LBP and body composition is well-documented. You et al. [35] suggest that an augment of 20% in body fat mass increases the risk of LBP. Also, Glänzel et al. [57] demonstrated that in rural workers with LBP, the sample group showed body composition deviation.

However, the results of the study were not wholly consistent with the statement that spinal curvatures are a determinant of self-reported disability. Perhaps the causes of disability include the feedback between obesity and low levels of physical activity. With these two variables, sedentary lifestyles and the respective sedentary body position are conducive to the involution of body posture and increased kyphosis. Therefore, the prevention of back pain should be supported by conscious and purposive activity aimed at compensating for motor deficits to change the habitual position of individual body segments. Adequately chosen specific or directional movements are recommended when the patient already suffers from pain syndrome, whereas general fitness exercises play a preventive role. The European Commission recommends such activities in their guidelines for acute and chronic LBP syndromes (Working Group B13 European Cooperation in the Field of Scientific and Technical Research—COST) [58].

Study limitations

Although this study aimed to identify the relationships and impacts of specific selected variables among older women, an important focus should be determining cause-and-effect relationships. Therefore, we recommend further research that explicitly aims to identify these causes. Also, future studies should consider a more comprehensive assessment of the potential confounding factors of the elderly population, which we consider a limitation of our study. Even though we made efforts to standardize data collection and minimize bias, factors such as hormonal changes, bone density, muscle strength, previous injuries, or lifestyle factors may introduce variability that could affect the observed associations. The last point is about the low R^2^ values that indicate limited explanatory power of the regression models, which may reduce the generalizability of the results.

## 5. Conclusions

This study did not establish a direct correlation between low back pain syndromes and spinal curvatures in the sagittal plane. Only these three factors (PA, body build, and body composition), in conjunction with age, contributed to a statistically significant risk of experiencing pain. To address these issues, educational initiatives should be implemented for older adults. These programs should promote specific, purposeful, and tailored physical activities to enhance health, encourage ergonomic lifestyles, and help maintain proper body posture during work and leisure activities. Preventive measures should also focus on avoiding static body positions, managing weight, and teaching correct techniques for lifting and moving heavy objects.

## Figures and Tables

**Figure 1 healthcare-13-01054-f001:**
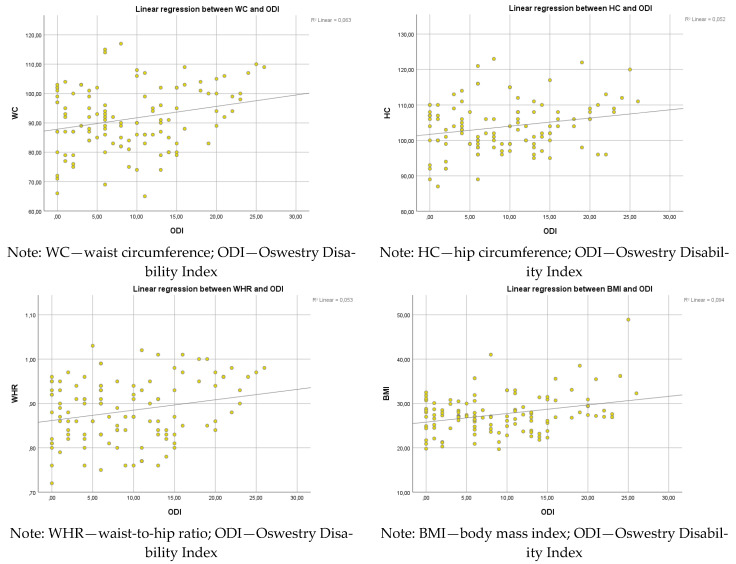
Linear regression between ODI and WC, HC, WHR, and BMI.

**Figure 2 healthcare-13-01054-f002:**
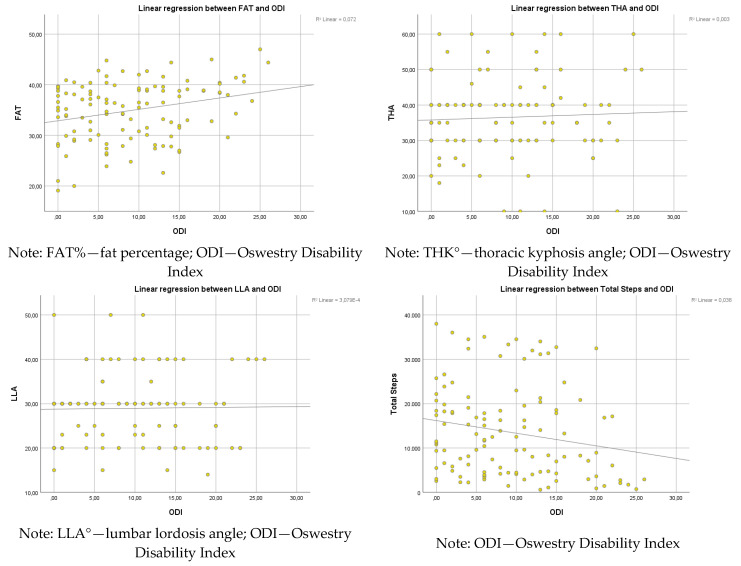
Linear regression between ODI and FAT, THK, LLA, and total steps.

**Table 1 healthcare-13-01054-t001:** Characteristics of the physical status of the study participants.

Variable	N (Valid)	Mean ± SD	Median	Min–Max
WC (cm)	114	91.38 ± 10.81	91.00	65.00–117.00
HC (cm)	114	103.79 ± 7.03	104.00	87.00–123.00
WHR	114	0.88 ± 0.07	0.88	0.72–1.03
BMI	114	27.57 ± 4.42	27.10	19.70–48.90
FAT%	114	34.94 ± 5.83	36.30	19.10–47.00
THK°	114	36.50 ± 10.66	35.00	10.00–60.00
LLA°	114	28.95 ± 7.56	30.00	14.00–50.00
Total steps	114	13,599.86 ± 10194,66	11,486.00	582.00–38,008.00
ODI	114	9.08 ± 6.97	8.00	0.00–26.00

Legend: WC—waist circumference, HC—hip circumference, WHR—waist-to-hip ratio, BMI—body mass index, FAT%—fat percentage, THK°—thoracic kyphosis angle, LLA°—lumbar lordosis angle, ODI—Oswestry Disability Index.

**Table 2 healthcare-13-01054-t002:** The Oswestry Disability Index and outcome variables.

Outcome Variable	F-Statistic	*p*-Value	R^2^ Value
WC (cm)	F(1, 113) = 7.574	*p* = 0.007	R^2^ = 0.063
HC (cm)	F(1, 113) = 6.257	*p* = 0.014	R^2^ = 0.052
WHR	F(1, 113) = 6.342	*p* = 0.013	R^2^ = 0.053
BMI	F(1, 113) = 11.660	*p* = 0.001	R^2^ = 0.094
FAT%	F(1, 113) = 8.806	*p* = 0.004	R^2^ = 0.072
THK°	F(1, 113) = 0.290	*p* = 0.591	R^2^ = 0.003
LLA°	F(1, 113) = 0.290	*p* = 0.591	R^2^ = 0.003
Total steps	F(1, 113) = 4.446	*p* = 0.037	R^2^ = 0.038

Legend: WC—waist circumference, HC—hip circumference, WHR—waist-to-hip ratio, BMI—body mass index, FAT%—fat percentage, THK°—thoracic kyphosis angle, LLA°—lumbar lordosis angle.

## Data Availability

The original contributions presented in this study are included in this article; further inquiries can be directed to the corresponding author. The data that support the findings of this study are available on request from the corresponding author. The data are not publicly available due to privacy or ethical restrictions.

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
