# Peer review of "Exploring the Relationship Between Low Back Pain, Physical Activity, Posture, and Body Composition in Older Women"

_healthcare, 2025, doi:10.3390/healthcare13091054_

Round 1
Reviewer 1 Report
Comments and Suggestions for Authors
This manuscript addresses a relevant and timely topic in public health and gerontology, exploring the associations between LBP, PA, posture, and body composition in older women. While the study concept is valuable, the overall research design, statistical approach, and interpretation of findings contain several methodological and structural flaws. These significantly compromise the validity and scientific contribution of the manuscript. Based on the current form, I recommend major revision with the possibility of rejection unless fundamental issues are adequately addressed.
Major Concerns:
- Methodological Issues:
- Study Design Mismatch: The study uses a cross-sectional design, yet draws inferences suggestive of prediction and causality (e.g., “predictors of LBP”), which is methodologically inappropriate. The language used throughout the manuscript (e.g., “influence,” “predict”) misrepresents the nature of the data and should be corrected.
- Lack of Conceptual Framework: The manuscript lacks a clear theoretical basis for selecting and analyzing variables. There is no reference to a guiding model (e.g., the biopsychosocial model of pain), which could provide coherence to the hypothesized associations.
- Confounding Factors Omitted: Important potential confounders—such as comorbidities, medication use, pain chronicity, physical activity history, or psychosocial variables—were not included or adjusted for, weakening the internal validity of the findings.
- Inadequate Statistical Modeling: The authors primarily rely on simple linear regressions for each variable without constructing a multivariate model. This ignores potential multicollinearity and interactions between variables. Moreover, the reported R² values are very low, indicating poor explanatory power, yet the findings are overstated.
- Sampling Representativeness Not Addressed: The sample is drawn from a university for older adults (University of the Third Age), likely representing a healthier and more active subset. The manuscript fails to discuss how this may bias the generalizability of the findings.
- Interpretation and Discussion:
- Overstatement of Results: Several variables (e.g., BMI, waist circumference) are reported as “strong predictors” of disability scores, despite R² values of less than 0.1, indicating weak practical significance.
- Shallow Analysis: The discussion lacks depth and critical interpretation. Results are merely compared to previous studies without any theoretical synthesis, exploration of mechanisms, or discussion of conflicting findings.
- Contradictory Emphasis: Postural measures (e.g., THA and LLA) were found to be non-significant, yet are still given considerable attention in the discussion without adequate justification.
- Presentation and Structural Issues:
- Inconsistencies in Style and Formatting: The manuscript contains inconsistent use of abbreviations, non-standard typographic formatting (e.g., font inconsistencies in tables), and numerous grammatical issues that detract from readability and professionalism. Careful proofreading and copyediting are needed.
- Poor Data Visualization: Figures and tables lack appropriate labels, units of measurement, and confidence intervals. Figures are not informative and fail to effectively support the statistical claims.
Recommendations for Revision:
To improve the manuscript, the authors should:
- Avoid causal language and accurately frame findings as correlational.
- Include multivariate analyses to control for confounding variables.
- Adopt a theoretical framework to support variable selection and interpretation.
- Differentiate clearly between statistical significance and clinical relevance.
- Revise figures/tables for clarity, accuracy, and consistency.
- Improve writing style for precision, grammar, and formatting throughout.
- Enhance discussion depth by integrating theoretical perspectives and exploring underlying mechanisms.
- Acknowledge sample bias and discuss generalizability.
Author Response
Dear reviewer,
Thank you for taking your time to review our manuscript. We attached a PDF document with the answers to your requests. If you need further clarifications, we ar at your disposal.
Thank you!
The authors.

Reviewer 2 Report
Comments and Suggestions for Authors
I think could be important to considerer the following:
A/. Why didn't you use the Kolmogorov-Smirnov test (KS) to check the data's normality assumption ? Theoretically, KS test is more appropiate when the n > 50.
B/. In tables 1,2, why don't you add variable's units: Angles º, cm /", kgs/% fat, etc
C/. Where did you meaure the Waist Circumference (WC): it was the minimum perimeter/girth, at the umbilicus level, etc ?
D/. For a good health, more important than Total Tat (TF) is Total muscle.
E/. There's an international consens about the importance of the Erector Spinalis muscles' strength to prevent LBP. Would be very interesting to measure it.
F/. Despite your study didn't find a relationship between the ODI and postural deviation (thoracic kiphosis and lumbar lordosis), you should explore the international bibliography about it...
G/. Last but not least. despite the "n" (114) of your study is acceptable, in a cross-sectional study of this type, isn't good enough to find explanatory powers of the regression models.
Author Response

(The authors gave the same response as above.)

Reviewer 3 Report
Comments and Suggestions for Authors
Thank you for the opportunity to review this manuscript, which considers some interesting, applied issues.
This study appears to be novel, and author showed an interesting point about “Exploring the relationship between low back pain, physical activity, posture and body composition in older women”.
The manuscript is written in accordance with the journal's guidelines, contains an adequate paper structure, and all chapters are very concise and easy to read.
Based on what I have read, I notice a few things that would be good to correct, in order to improve the quality of the article.
Please see my comments below.
ABSTRACT
Line 18- Move the information about the sample of subjects to the methods section.
Line 21-23 - Add the numerical values ​​of the results
Reorganize this section to provide readers with more concise and precise information about what they can find in your manuscript.
Edit the keywords because they are all already contained in the title
INTRODUCTION
The introduction is too short and does not contain all the necessary information about the topic of the manuscript. Reorganize this section and try to make your paragraphs contain more than one sentence. Also, references are missing after some sentences.
METHODS
The paper does not offer enough information about the size and diversity of the sample of respondents, which may affect the generalization of the results.
I ask the authors to clearly state when the study was conducted, given the fact that the research protocol was approved by the Bioethics Committee in Katowice, Poland (No. 9 with addendum KB/29/2020) with an apparent date of 2020?
When you first list the variables used in this paper, maintain the order throughout the rest of the manuscript.
Add a reference for anthropometric parameters and write where the instrument has been used so far. For each test you used, add where it was previously used, as well as the manufacturer for the instruments.
DISCUSSION
At the beginning of the discussion, clearly let readers know what the main findings of your study are. Then later write about what other authors have found in their papers.
Although the results are clearly presented, their interpretation in the context of earlier research could be more detailed. I suggest the authors to add a deeper comparison with previous research in order to better emphasize the novelty of this work.
I urge the authors to conduct a more comprehensive review of the recent literature and integrate studies published within the last five years. This will not only strengthen the theoretical framework and discussion but also enhance the manuscript’s overall impact and reliability.
Also, it is necessary to bold the year of publication in accordance with the instructions for authors in MDPI journals.
Edit Author Contributions according to the instructions for authors, correcting to initials.
Author Response

(The authors gave the same response as above.)

Reviewer 4 Report
Comments and Suggestions for Authors
First, I would like to thank the authors for their work and the editor for the opportunity to revise the manuscript entitled Exploring the relationship between low back pain, physical activity, posture and body composition in older women.
The manuscript fits within the journal’s aims and scopes. However, I have some concerns, and some points deserve modifications and suggestions for improving the manuscript's general appearance.
There are some old references that do not seem to be method articles, two 25-year-olds and one older. Are the list of articles below still reliable, or are there newer references that can be used?
Article number 32 is from 1999
Articles number 13 and 22 are from 2000.
Article 14 is from 2002.
Article 31 is from 2004.
Articles 26, 29 and 30 are from 2005
Article 19 is from 2006.
Articles 25 and 29 are from 2009.
Articles 23 and 24 are from 2010.
Articles 15 and 18 are from 2013.
There is one reference without a Year reference (number 21).
There are two references where the titles are not translated into English (numbers 19 and 23).
The authors may want to replace some articles with newer ones as conditions change over time, for example, with newer treatments, and make the present article more reliable.
Abstract:
When the study population is described in the background, I miss the area where the study took place (Page 1, line 18).
Introduction:
In the introduction, I miss a short description of physical activity and the WHO recommendations for older adults. I also miss short description of body composition and recommendations.
Materials and Methods.
When the study population is described, I miss the area where the study took place (Page 2, lines 72-73).
I also wonder why the authors refer to the Declarations of Helsinki 2013 when there is an updated version of 2024. (Page 2, line 83)
Result:
In the method section, some abbreviations are described such as WHR, BMI, WC HC and some more but in the result they are not used consistently. For example in page 3, lines 124-128.
I suggest that the authors are consequent.
The figures should be able to read without the other text, which means that the abbreviations must be explained in the figure texts. Figures 1 and 2.
Discussion:
The majority of the articles in the discussion are old, which could affect the reliability of the present article.
Conclusion:
The conclusions are supported by the result.
Author Response
Dear reviewer,
Thank you for taking your time to review our manuscript. We attached a PDF document with the answers to your requests. If you need further clarifications, we are at your disposal.
Thank you!
The authors.

Round 2
Reviewer 1 Report
Comments and Suggestions for Authors
The revised manuscript has been systematically revised to address the issues raised, and some of the research design and methodological issues have been clarified in the limitations, and is recommended for acceptance with minor revisions in view of the overall quality of the revisions and the potential contribution of the paper to the field. The minor revisions include, please double-check that the references match the journal's format, some minor grammatical errors. Also, please note that the section on study limitations is generally placed at the end of the discussion section rather than the conclusion section.
Author Response
Dear reviewer,
We attached the new responses for your suggestions.
Thank you for helping us to improve the quality of our manuscript!
The authors
